# Instance Generation for Maximum Independent Set using Graph Generative Models

**Iker Pérez**[1]**, Josu Ceberio**[2]**, Iñigo Urteaga**[3]
[1]UPV/EHU, `iperez267@ikasle.ehu.eus`
[2]UPV/EHU, `josu.ceberio@ehu.eus`
[3]BCAM, `iurteaga@bcamath.org`

## Abstract

In the field of combinatorial optimization, having problem instances is essential for the design, development, and evaluation of algorithms and models. However, their availability is usually limited and, in general, existing benchmark repositories are used to carry out the above-mentioned tasks along with current combinatorial optimization problem (COP) instance generators. The problem is that those instances are usually artificially generated and do not always succeed in reflecting real problem properties, such as a the solutions achieved by a given algorithm. In this paper, we approach the COP instance generation development using graph generative models. Specifically, given a reference sample of instances of a given problem, the aim is to implement a model that is able to generate (i.e., sample) new instances from the probability distribution on the COP instance space that mimics the original examples. In this particular case, we assume that the instances of optimization problems we are interested in can be represented as undirected unweighted graphs. Because of that, we will focus on the maximum independent set problem. Conducted experiments show that although we are capable of creating graphs similar to the original ones, their properties do not coincide with the expected ones, making room for other models and approaches.

## 1  Introduction

Combinatorial Optimization Problems (COP) have recurrently been studied by the communities of operational research, applied mathematics, and computer science, producing a wide variety of works that propose exact [PR91, Cpl17, PF], heuristic, meta-heuristic [BR03] and, more recently, machine learning strategies [BLP21, BPL+16] in order to solve them.

Generally, developing an algorithm that works better than the rest for a given problem, and therefore, for all possible instances of that problem, is not possible without contravening the No-Free-Lunch Theorem [WM97]. For this reason, the interest lies in developing algorithms with the desired characteristics (better in performance, better in efficiency, better in complexity, etc.) for a specific type of instances. *Instance* is understood as a set of values that parameterize the objective function of the problem to be optimized, but that does not change its mathematical formulation. In other words, it could also be understood as an example of the problem being represented. In the case of the Maximum Independent Set (MIS), an instance would be represented by any undirected unweighted graph $G = (V, E)$ of $|V| = N$ nodes. As mathematically described in Equation 1, MIS is the problem of finding a set of nodes $x \subseteq V$, such that no two vertices in $x$ are adjacent and the cardinality of $x$ is maximized. If the first condition is satisfied, the set is considered independent, hence the name. In the mathematical representation of MIS, instances are defined with the adjacency matrix $D = [d_{ij}]_{N \times N}$, and the set to be evaluated is represented as $x$.

$$X^* = \arg\max_{x} \begin{cases} |x| & \text{if } \sum_{i=1}^{|x|} \sum_{j=i+1}^{|x|} d_{x(i)x(j)} = 0 \\ 0 & \text{otherwise} \end{cases} \tag{1}$$

XVI XVI Congreso Español de Metaheurísticas, Algoritmos Evolutivos y Bioinspirados (maeb 2025).

The development of the algorithms to solve MIS and other COP problems require a significant number of instances that are needed to execute experimental comparisons and, in certain cases, carry out costly learning processes where thousands of instances are necessary. In contrast, the availability of optimization instances rarely exceeds a hundred, especially in problems from real environments. As a result, the community has opted to use existing repositories of instances (commonly called benchmarks) and has combined them with instances generated artificially by sampling their parameters from certain probability distributions (usually uniformly at random) [DLM11].

In the absence of alternatives, the artificially generated instances have allowed for the entire development of the field; however, it is unknown what the actual performance of the algorithms would be if more realistic instances were used for the development of algorithms, and in the same way, if models generating instances with properties firmly matching those from real world were used. As Ishibuchi et al. [IHS19] noted, the performance of an algorithm on popular benchmark problems can be different from that on real-world problems. Tanabe et al. [TO17] also show that widely-used real-world-like problems have some unnatural problem features, so the necessity of a large amount of real-world problems exists. The problem lies in the fact that it is difficult to implement generators that create instances that are similar to the real ones, since it implies knowledge about how instances occur in the real-world and about their inner structure, which is as difficult as solving them [CHML13].

Despite unsuccessful attempts to generate realistic instances in the literature of COPs, relevant progress has been made in the field of generative modeling, where structured data from numerous domains (audio, video, images, graph, etc.) is being artificially generated using a sample of reference data. Those generative models have shown a high capacity to learn the inner structure of the data they have to represent. In the particular context of COPs, many of these problems can be naturally described as graphs, therefore, we have formulated the generation of such instances as a graph generation task. However, as graphs are not universally suitable for representing all type of COPs, for problems where alternative representations are more appropriate, correspondingly tailored representations and architectures should be used.

Among graph generative models, we will investigate deep learning-based models due to their demonstrated superior performance in other tasks, which we hypothesize might be relevant for COP instance generation as well. As our work here is a first step in that direction, we include a thorough review of graph generative models that could also be used in future work. We emphasize those that have interesting attributes for COP instance generation (permutation invariance, size invariance and capacity for large graph generation) and classify them based on the type of graph they generate. We also present a first approach to build COP instances using graph generative networks. Specifically, we take MIS as the case of study and implement GraphRNN [YYR+18] to generate instances as similar as possible to the reference set. The experiments carried out reveal that it is feasible to use GraphRNN [YYR+18] to recreate MIS instances, although we also observed certain limitations in the generative capacity of the model.

This paper is organized as follows. Section 2 presents previous work on the area of COP instance generation and presents current models for graph generation in the literature. Additionally, we provide a table that briefly summarizes the key characteristics of these models in handling COP instance generation. Then, Section 3 details the proposed methodology, encompassing the model architecture, dataset preparation, training protocol and evaluation metrics. Afterwards, Section 4 presents a comprehensive analysis of experimental results. Finally, Section 5 concludes the paper with key findings and insights and outlines promising directions for future research.

## 2 Background

### 2.1 Previous work on generating instances for COPs

Creating instances of COPs with desired properties has been a long-standing research line. The most naive instance generators create instances by randomly sampling their parameters from user defined distributions (usually uniform or Gaussian) [DLM11]. In some other works, limited information about instance's parameters is used to feed the hyperparameters of the mentioned distribution. There have been some advanced works using these methods, such as the Max-Set of Gaussian Landscape Generator [GY06], or Kriging based real-like landscape generator [FZS+16].

In the first case, the Max-Set of Gaussian Landscape Generator computes the upper envelope of $m$ weighted Gaussian process realizations and can be used to generate continuous, bound-constrained optimization problems. The landscape generator is parameterized to control the features of the

instances to be generated. Among others, the parametrization can include terms like the number of Gaussian components and implicitly, the number of local optima.

The Kriging based real like landscape generator uses a Kriging model [SSK18] to fit to a real dataset, then this model is altered according to a variation parameter $\alpha$, by adding it to the parameters of the model. The generated instances will be discarded or kept according to thresholds for several similarity measures of the model.

However, although instances obtained by those methods have been widely used in the combinatorial optimization area, it cannot be assumed that they are able to replicate real instances. Generating COP instance datasets according to a specific distribution is a non-trivial task. Ceberio et al. [CML17] show this difficulty by applying a reduction function ($f : N \times N \longrightarrow r$, where $r \in \mathbb{R}^+$) to the artificial instances. This function provides the solution for the problem represented by the instance, i.e., the solution of the combinatorial problem that is being represented. In that article, the authors note that when sampling the parameters uniformly at random, the problems analyzed in terms of rankings of solutions in the search spaces are not distributed uniformly. This results points a stray weakness of such a technique so commonly used.

## 2.2 Literature review of graph generative models

In order to analyze the capabilities of the available models for graph generation, we will mainly focus on four of their attributes: permutation invariance, size invariance, maximum graph size achieved and type of graph generated (weighted / unweighted, directed / undirected, ...). Due to the permutation invariant nature of COPs, the vast amount of equivalences that appear when representing the instances may transform the learning of its structure into a difficult task. This, combined with the limited availability of extensive datasets, may lead to low-quality generated instances. Size invariance along with maximum graph size achieved are also good attributes for COP generative models, as COPs can be of very distinct sizes, reaching up to thousands of nodes. A model with size invariance and ability to deal with large graphs makes it a highly versatile choice for working with different types of COPs. Finally, the type of graph generated is crucial as each COP has its own representation and not all types of graphs are suitable.

In the following, we provide a taxonomic classification of graph generative models, along with a table summarizing their main characteristics (Table 1). Four main families will be analyzed: models based on Variational AutoEncoders (VAE) [BL14], those based on flow models [RM15], on diffusion models [SDWMG15] and on autoregressive models like Recurrent Neural Networks (RNNs) [Elm90].

Among the earliest approaches in the literature, we find models based on the VAE architecture. The most representative model using this framework is GraphVAE [SK18], which decodes a vector representation into a fully connected probabilistic graph, through a one-shot process. The low capacities of VAEs to generate high quality graphs have led to their abandonment in favor of other alternatives. However, this architecture has recently been revived for its use in combination with diffusion models, achieving higher quality graphs. Among others, NGG [ENC+24] and PARD [ZDA24] are models that leverage this combination of architectures. In the case of the latter, its results surpass those of many existing reference models, such as EDP-GNN [NSS+20], DiGress [aSLH22] or GDSS [aSLH22].

Notable graph generators that implement flow-based models are GraphDF [LYJ21], GraphNVP [MINA19], CatFlow [EBN+24] and MoFlow [ZW20]. The last two groups, the ones using autoregressive and diffusion models, stand out with superior performance, thus significant popularity. The major contributions in the domain of autoregressive models, GraphRNN [YYR+18] and GRAN [LLS+19], despite not being permutation invariant, are notably simpler than those made with other architectures. Due to their regressiveness, these models can handle a broader range of graph sizes compared to others, highlighting their size invariance property. Both models, GraphRNN and GRAN, have appropriate features to deal well with COP problems as they reach large graph lengths, and in the case of GRAN, it could also be appropriate for a large variety of COPs.

Finally, diffusion models have become a very prolific architecture in terms of new proposed models in recent years. Those, although not being as simple as autoregressive ones, also obtain very good results, making them a promising area of research. Among others, we find LGGM [WRP+24], Twigs [MVFG24] and FairWire [KS24]. Some of the proposed diffusion models are permutation invariant,

which makes them very suitable for our purpose. Graph generators with such a property are: DisCo [XQC$^+$24], DiGress [aSLH22], GDSS [aSLH22] and EDP-GNN [NSS$^+$20].

Table 1: Summary of graph generative models for instance generation. Only articles published since 2018 were considered. Checkmarks in *Perm. inv.* column indicate that the model is able to deal with the permutation variability of the graphs naturally, and without involving costly processes. Checkmarks in *Size inv.* column indicate that the model is able to generate graphs of different sizes without changing its architecture. *max. $|V|$* column indicates the maximum graph size achieved in the original experimentation. *Code* column indicates whether the code is available. In the *Application* column we indicate which type of COP problem instances is the model able to generate indicating unsuitability if the generated graphs are too small. Finally, '-' identifies those cases where characteristic is not clear and red colour the ones a characteristic is believed to be present but cannot be assured.

| Model | Ref. | Year | Perm. inv. | Size inv. | max. $V$ | Code | Generated graph type | Application |
|---|---|---|---|---|---|---|---|---|
| DisCo | [XQC$^+$24] | 2024 | ✔ | ✔ | $|V| = 187$ | ✔ | Graphs with categorical node and edge attributes. | MIS, MDS, GPP, GCP. |
| Twigs | [MVFG24] | 2024 | - | ✔ | $|V| = 126$ | ✔ | Graphs with categorical edge attributes. | MIS, MDS, GPP, GCP. |
| PARD | [ZDA24] | 2024 | ✔ | ✔ | $|V| = 210$ | ✔ | Graphs with categorical edge attributes. | MIS, MDS, GPP, GCP. |
| FairWire | [KS24] | 2024 | ✘ | ✔ | $|V| >= 169$ | ✘ | Undirected unweighted graphs. | MIS, MDS, GPP, GCP. |
| CatFlow | [EBN$^+$24] | 2024 | ✔ | ✔ | $|V| = 210$ | ✘ | Graphs with categorical edge attributes. | MIS, MDS, GPP, GCP. |
| LGGM | [WRP$^+$24] | 2024 | ✘ | ✔ | $|V| > 265$ | ✔ | Undirected unweighted graphs. | MIS, MDS, GPP, GCP. |
| NGG | [ENC$^+$24] | 2024 | ✔ | ✔ | $|V| = 100$ | ✔ | Undirected weighted graphs. | LOP, TSP, MIS, MDS, GPP, GCP |
| DiGress | [VKS$^+$22] | 2023 | ✔ | ✔ | $|V| = 200$ | ✔ | Graphs with categorical node and edge attributes. | MIS, MDS, GPP, GCP. |
| GDSS | [aSLH22] | 2022 | ✔ | ✔ | $|V| = 400$ | ✔ | Graphs with categorical node and edge attributes. | MIS, MDS, GPP, GCP. |
| GraphDF | [LYJ21] | 2021 | - | - | $|V| = 38$ | ✘ | Graphs with categorical node and edge attributes. | Not suitable for our purposes. |
| EDP-GNN | [NSS$^+$20] | 2020 | ✔ | ✔ | $|V| = 20$ | ✘ | Undirected weighted graphs. | Not suitable for our purposes. |
| MoFlow | [ZW20] | 2020 | - | - | $|V| = 38$ | ✔ | Graphs with categorical node and edge attributes. | Not suitable for our purposes. |
| GraphNVP | [MINA19] | 2020 | ✘ | ✔ | $|V| = 38$ | ✔ | Undirected unweighted graphs, with attributed nodes. | Not suitable for our purposes. |
| GRAN | [LLS$^+$19] | 2019 | ✔ | ✔ | $|V| = 5037$ | ✔ | Undirected unweighted (and possibly weighted) graphs. | LOP, TSP, MIS, MDS, GPP, GCP. |
| GraphRNN | [YYR$^+$18] | 2018 | ✘ | ✔ | $|V| = 500$ | ✔ | Undirected unweighted graphs. | MIS, MDS, GPP, GCP. |
| GraphModel | [LVD$^+$18] | 2018 | ✘ | ✔ | $|V| = 20$ | ✘ | Undirected weighted graphs. | Not suitable for our purposes. |
| GraphVAE | [SK18] | 2018 | ✘ | ✘ | $|V| = 38$ | ✔ | Undirected weighted graphs. | Not suitable for our purposes. |
| Survey1 | [ZDW$^+$22] | 2022 | | | | | | |

## 2.3 Graph generative models for COP benchmark generation

Not all graph generative models are suitable for generating COP benchmarks. We study GraphRNN [YYR$^+$18], LGGM [WRP$^+$24] and GRAN [LLS$^+$19] closely, based on their properties, which we consider most suitable for our task. The first model, which is a reference in graph generation, has been used as baseline for comparisons in most of the follow-up works described in Table 1. The original article states that it can replicate graphs with up to 500 nodes, showing potential flexibility with different problem sizes. On the other hand, the strong point of LGGM is that it does not require a large number of instances to generate high-quality outputs, as it is pre-trained on a substantial dataset of graphs. Due to the type of graphs they model, undirected unweighted graphs, GraphRNN and LGGM can be used to generate instances of MIS, Graph Partitioning Problem (GPP), Graph Coloring Problem (GCP) and Minimum Dominating Set (MDS) among others. GRAN also shows a very good potential in this field, as it can generate substantially larger graphs than the other models, capture correlations between generated edges and furthermore it could be suitable for Linear Ordering Problem (LOP) or Traveling Salesman Problem (TSP) like COP problems, where edges between nodes are weighted, in addition to the ones previously mentioned. For this study on COP

instance generation, GraphRNN was selected as the experimental model, based on its recognition as a reference in graph generation, its capacity to generate large graphs and its comparatively less complex architectural structure.

## 3   Experimental Setting

This section aims to outline the procedure for rigorously validating the selected graph generative model, GraphRNN, in generating high-quality MIS instances.

### 3.1   Model

GraphRNN consists of a node-level recurrent neural network that sequentially generates nodes while maintaining a hidden state encoding the structure generated so far. It represents a hierarchical approach to graph generation, where the graph's structure is decomposed into sequences that capture both node-level and edge-level dependencies. The model employs two types of RNNs: a graph-level RNN that maintains the state of the graph generation process and generates new nodes, and an edge-level RNN that creates edges for each new node by predicting its connections to previously generated nodes.

The process of graph generation can be seen in Figure 1, where the graph-level RNN functions as the master controller, maintaining a graph-level hidden state that evolves as the generation progresses. For each time step, it decides whether to add a new node to the graph. When a new node is added, the edge-level RNN is activated to generate the node's connectivity pattern. This edge-level RNN operates on a sequence of binary decisions, determining for each previously generated node whether an edge should exist between it and the new node, effectively capturing complex edge dependencies.

The training process learns from true graph sequences obtained through a breadth-first-search (BFS) ordering of nodes from the training graphs. This ordering provides a consistent representation of graphs and helps the model learn meaningful patterns in graph structure. The model is trained to minimize the negative log-likelihood of the proposed generative model over observed node and edge sequences, allowing it to capture both local and global graph statistics while maintaining computational efficiency through its sequential generation approach.

Figure 1: GraphRNN at inference time. Green arrows denote the graph-level RNN that encodes the "graph state" vector $hi$ in its hidden state, updated by the predicted adjacency vector $S_i^\pi$ for node $\pi(vi)$. Blue arrows represent the edge-level RNN, whose hidden state is initialized by the graph-level RNN, that is used to predict the adjacency vector $S_i^\pi$ for node $\pi(vi)$. Figure obtained from GraphRNN original paper [YYR+18].

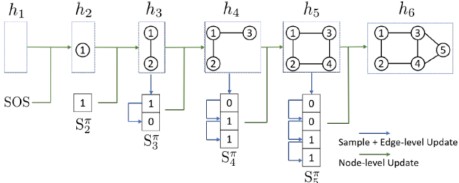

### 3.2   Dataset

To measure the power of GraphRNN when generating MIS instances, we will use two dataset approaches. On the one hand, artificial instances, which follow a known distribution and, on the other hand, real instances, which follow an unknown distribution. With that purpose, three different datasets have been used: an artificial dataset made of graphs created with the Erdös-Rènyi model [ER+60] (which was chosen because, despite producing artificial graphs, it offers a cost-effective way to generate graphs) and two realistic datasets derived from graphs in DIMACS repositories. By "realistic", we refer to graphs that exhibit properties comparable to those of real-world graphs.

To address the lack of large, equally sized, and distributed benchmarks of MIS instances required for training the model, we have developed custom, realistic datasets by adapting real-world graphs used in combinatorial optimization: a graph showing the internet topology [LKF05] and a graph showing the connectivity of New York roads [RA15]. In order to best approximate real graph distributions, we have constructed the new graphs by making a sampling from the edges and nodes of the original ones. For doing that, we have created non-overlapping partitions to define the subgraphs. Let $G = (V, E)$

be the original graph, where $V$ is the set of vertices and $E$ is the set of edges. For all child graphs $G_i = (V_i, E_i)$ extracted from the original one:

$$\forall i \longrightarrow (V_i \subset V \text{ and } E_i \subset E), \quad \forall i \neq j \longrightarrow (V_i \cap V_j = \emptyset \text{ and } E_i \cap E_j = \emptyset).$$

To ensure similar distributions between the training and test datasets, the test graphs have been created using the same nodes and edges as those in the training set. However, to prevent identical graphs from appearing in both partitions, each test graph has been constructed by combining nodes from two different training graphs. When creating the dataset, the length of the instances has been set to 100 nodes and the length of the dataset to 500 instances. The intention is to test if GraphRNN is able to replicate MIS instances rather than to prove if it can do it in a truly realistic scenario. Thus, in those experiments dataset size is not a constraint.

### 3.3 Training procedure and evaluation metrics

A training of 8000 epochs has been proposed to ensure a consistent training procedure across datasets and training configurations (seeds, learning rates, etc.). To evaluate the stochasticity of the learning process, we replicated the experiments 16 times, employing two distinct learning rates ($10^{-5}$ and $10^{-6}$) and eight different seeds.

The evaluation of the generated instances is crucial to prove the capacity of our model. As our combinatorial problems are represented as graphs, part of our evaluation process must be similar to the evaluation of graph generative models. To this end, degree and clustering metrics [New03], standard in the field of graph generation models, will enable us to determine if the generated graphs exhibit sufficient similarity to the original graphs from a structural perspective. Furthermore, we will employ four metrics from the combinatorial optimization field to gain deeper insights into the structural properties of the instances. Solving the MIS instance for both original and artificial datasets will allow us to compare the resulting solutions and thereby extract information about the instance's inherent properties. Solutions of the problem represented by the instances will be compared by means of their distribution using both Maximum Mean Discrepancy (MMD) and Kullback–Leibler divergence (KL) [Kul51].

In order to compare the solutions of the problems represented by the dataset and those represented by generated graphs, we will first solve the problems using two different methods: simulated annealing [BT93] and estimation of distribution algorithm [LL12]. These algorithms, each employing a different optimization strategy, enable us to detect solution-based discrepancies between original and generated instances, leveraging heuristic nature of the algorithms to reveal variations.

## 4 Experimental results

To interpret the results, we will first analyze information provided by the metrics and then examine the training and validation losses, as these also provide crucial insights into model performance.

### 4.1 Results in terms of metrics

In Table 2 three groups of three rows can be observed. Each group is related to a different dataset (Erdös-Rènyi, NY roads and Internet Topology). Within each group, the rows are organized as follows: the first row represents the results of comparing the original training dataset with graphs obtained from the same distribution as those, thus indicates expected performance. The second row shows results of graphs that the model outputs with a learning rate of $10^{-5}$ and the third row shows results from graphs generated after a learning rate of $10^{-6}$.

When referring to the columns, the first and second columns represent the MMD for degree and clustering coefficients between original and predicted graphs. SA/EDA column shows the MMD between the results obtained when evaluating the instances with simulated annealing and estimation of distribution algorithm. Although being two different experiments both have been shown in the same column as results do not differ. Finally, SA/EDA KL columns show the Kullback–Leibler divergence between the distributions of results obtained when evaluating the instances with simulated annealing and estimation of distribution algorithm respectively. As in the previous column, the results of both experiments are shown together. With these metrics we can assess both, whether the graph structure and the representing COP instances are similar to the original ones. For all the presented metrics, a lower value indicates higher similarity of the generated graphs towards the original dataset.

Table 2: Results for different datasets

| Comparation dataset | degree | clustering | SA/EDA | SA/EDA KL |
|---|---|---|---|---|
| Erdös-Rènyi | 0.00060 | 0.00003 | 2.00000e+00 | 6.40879e-03 |
| GraphRNN (Erdös-Rènyi, $lr = 10^{-5}$) | 0.20072 | 0.00478 | 2.00000e+00 | 2.24628e-02 |
| GraphRNN (Erdös-Rènyi)($lr = 10^{-6}$) | 0.18862 | 0.00469 | 2.00000e+00 | 2.24628e-02 |
| NY roads | 0.04184 | 0.00121 | 1.99998e+00 | 4.79131e-01 |
| GraphRNN (NY roads)($lr = 10^{-5}$) | 0.09883 | 0.10717 | 2.00000e+00 | 5.53358e-02 |
| GraphRNN (NY roads, $lr = 10^{-6}$) | 0.09796 | 0.10281 | 2.00000e+00 | 5.53358e-02 |
| Internet Topology | 0.32786 | 0.03392 | 2.00000e+00 | 6.16101e-01 |
| GraphRNN (Internet Topology, $lr = 10^{-5}$) | 0.28604 | 0.09559 | 2.00000e+00 | 3.50549e-01 |
| GraphRNN (Internet Topology)($lr = 10^{-6}$) | 0.25500 | 0.06731 | 2.00000e+00 | 3.50549e-01 |

Our empirical evaluation indicates that there are differences in model performance across different training datasets. Table 2 shows the results of replicating Erdös-Rènyi distribution, where we see that the inner structure of the graphs in terms of degree and clustering differs substantially from their original counterparts. However, SA/EDA column has not captured any difference in terms of MMD and in the case of Kullback–Leibler divergence the difference has been minimum, as well. Therefore, it can be stated that although the generated problems were similar to the original ones in terms of their solutions, as the graph structure was not successfully replicated, the objective was not fully fulfilled.

Observing results related to NY roads dataset we see a similar pattern. The model is not able to completely replicate the graphs' inner structure, as degree and especially clustering differences are bigger than the reference ones. Nevertheless, metrics that involve the objective function of the COP instance, although being still different from the reference values, do not differ too much. Finally, results obtained with dataset Internet Topology are the ones that show the best solution to the instance generation problem, as the generated instances are clearly similar to the original ones in terms of both graph structure and COP instance representation metrics.

### 4.2 Training and validation loss results

Loss curves can be helpful to identify the relationship between graph quality and model performance. Internet Topology dataset, which achieved the best results, also had the lowest loss (see Figure 2). In contrast, the losses for the other two datasets did not fall below 0.28, leading to poorer performance, especially when analyzing the evaluation of the resulting COP instances.

The inability of the loss to descend below 0.25 is likely due to the model reaching its capacity limits. As observed, the training and validation losses follow nearly identical trajectories, indicating no significant overfitting. Overfitting is a common issue in machine learning, where a model performs well on training data but fails to generalize to unseen validation or test data [RM19]. In this case, however, the similar trends in training and validation losses suggest that the model's limitations stem from its architecture rather than overfitting, which is often associated with insufficient model complexity [GBCB16].

## 5   Conclusion and future work

The preliminary work presented in this paper notes that using graph generative models to create combinatorial optimization instances might be feasible, showing its potential to create real-world like problems. The model used here is understood as a first approach, as the goal of the work was to show the possibility of generating COP artificial instances using graph generative networks. Our results confirm that learning the underlying distribution of these instances is possible. While the chosen model may not be optimal, the experimental findings indicate that we can successfully generate MIS problems with a distribution similar to that of the input data.

However, there are significant challenges to be addressed. As it was observed in our results, the model improvement gets stuck when reaching a certain number of epochs. Although changes in the learning rate have been made, the loss gets the same value, which suggests that the model has reached a plateau in performance, where additional training does not yield better outcomes. This issue, though not fully understood, likely arises from GraphRNN's failure to capture these relations, as shown by the dissimilarity between generated and original graphs.

Figure 2: Training and validation loss evolution when using distinct datasets and learning rates

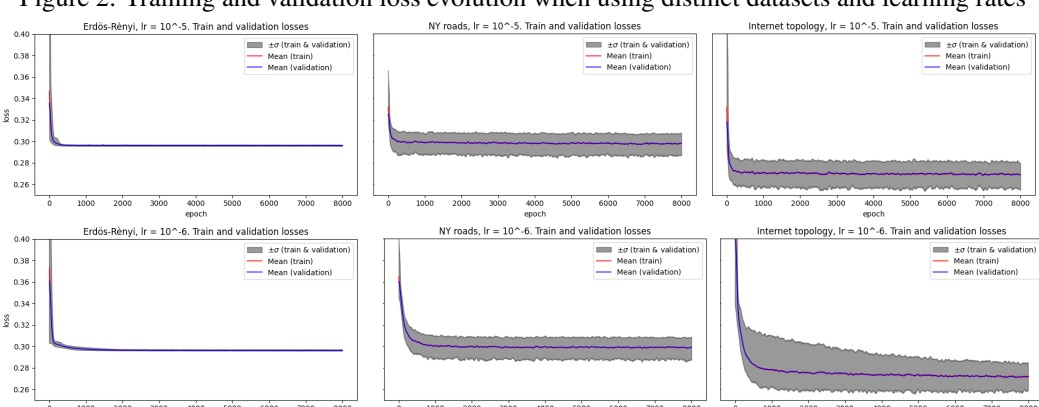

In future research, we aim to explore models specifically tailored for optimization instances. Currently, most models are designed for graphs and inherently account for both, edges and nodes. However, for COPs, it may be beneficial to develop models that do not rely on nodes and instead utilize edges more effectively to capture problem-specific features.

Another promising direction is to explore models like LGGM [WRP+24] and GRAN [LLS+19]. As shown in Section 2.4, these models could offer greater flexibility and generalization across various combinatorial optimization scenarios, making them potentially viable for COP instance generation.

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
