# OpenReview forum: "Instance Generation for Maximum Independent Set using Graph Generative Networks"
_MAEB/2025/Congreso — MAEB 2025_

### Official Review · Reviewer_YKCQ · 2025-03-11
**Instance Generation for MIS using Graph Generative Networks**

**Rating:** 5
**Confidence:** 4

**Review:**

The paper presents an initial approach on the generation of Combinatorial Optimisation Problem (COP) instances using Graph Generative Models (GGM). The authors aim to produce a model able to replicate the inner structure and features of real-world instances to improve the design and evaluation of algorithms. These are usually tested with benchmarking or randomly generated instances that may present different characteristics to real-world instances. Based on the assumption that many COPs can be represented as an undirected graph, they evaluate the ability of GraphRNN to generate similar instances for the Maximum Independent Set (MIS) based solely on the instance representation without any human feature extraction process.

The paper is well structured and presents not only a comprehensive review of the state-of-the-art in terms of graph generative models but also a well-conducted experimental evaluation. I consider that this paper should be accepted based on:

- It is reasonably argued that the need for real-world instances to design and evaluate algorithms.
- The authors effectively present the pros and cons of the different generation models before deciding on the chosen model.
- The use of three datasets with synthetic and ‘realistic’ instances of real-world graphs alongside the experimental design supports the results obtained and conclusions derived from them.

However, several issues may need to be discussed or corrected in the manuscript:

- The authors claim that many COPs can be naturally described as graphs (line 59) and while it is completely true, I wonder if using such representation may incorporate unnecessary complexity to some COPs. Although the graph-based representation is completely natural to the MIS problem, other COPs may benefit from tabular or text representations and the use of other generation models or strategies such as Generative Adversarial Networks (GAN).
- Sections 3.3 and 3.4 should be merged into a single section.
- Tables 2,3, and 4 may be simplified into a single table to save some space by grouping the columns SA and EDA into SA/EDA and columns SA-KL and EDA-KL into SA/EDA-KL and combining the rows for each dataset.
- Figure 2 could benefit from sharing the y-axis on each row and better colouring for the lines.
- In line 267 “(see Figure 2) but it did also incur in some overfitting, as the training loss gets under validation loss”, it is really difficult to appreciate that in the current figure.
- I do not agree with the definition of ‘overfitting’ in line 274 "Overfitting is a problem concerning many models in machine learning, and happens when models are able to discriminate too low level details.” A better suitable definition would be:

    > A problem in machine learning where a model performs well on the training data but it does not generalize will on the validation and/or test data.
    >
    >
    > Aurelien Geron. 2019. Hands-On Machine Learning with Scikit-Learn, Keras, and TensorFlow: Concepts, Tools, and Techniques to Build Intelligent Systems (2nd. ed.). O'Reilly Media, Inc.
    >

Finally, there are some typos that need to be fixed:

- Line 27: instances should start with capital I.
- Line 70: include a space between observed and certain.

---

### Official Review · Reviewer_Knfw · 2025-03-15
**Graph generative models to produce instances for the maximum independent set problem**

**Rating:** 5
**Confidence:** 4

**Review:**

The work presents the application of graph generative models that are trained to capture the inner structure of instances belonging to different datasets. Once the models have been trained, they are then used to produce new instances that aim to maintain the same internal structure that has been learned. The experimental evaluation focuses on the Maximum Independent Set (MIS) problem, a crucial combinatorial optimisation challenge.

The paper is very interesting and provides an original way to address a significant issue when working with combinatorial optimisation problems: the lack of instances. The paper is well-written and organised. While the experimental assessment is solid, the results did not align with the expected outcomes.

A comparison of the proposed generation models with those of other authors, such as K. Smith-Miles, J. Bossek or A. Marrero, would be a valuable exercise. Applying an Instance-Space Analysis (ISA) to all the different instances generated through the various methods will provide a clear picture of the particular location of your instances in the instance-space. This could help to better characterise your instances and obtain valuable information about the solvers used to deal with the generated instances. Some of the above methods require the specification of a set of features that define your MIS instances. I am not certain if this is the most appropriate approach for this particular problem domain, but there exist other alternatives that are feature-free.

Some minor typos:
- Line 36 -> "algorithms algorithms"
- Line 70 -> "observedcertain"

---

### Official Review · Reviewer_dbBw · 2025-03-19
**Interesante contribucion en el ámbito de los sistemas generativos para producción de bechmarks**

**Rating:** 4
**Confidence:** 4

**Review:**

Este estudio explora el uso de modelos generativos de grafos para la generación de instancias del problema del conjunto independiente máximo (MIS). Se implementa GraphRNN para generar instancias similares a las originales, con resultados prometedores pero con limitaciones en la reproducción precisa de las propiedades estructurales de los grafos.

En general, el trabajo me resulta correcto, si bien veo un poco forzada la relacion con el ámbito del congreso. En cualqueir caso, es un trabajo interesante. Algunos comentarios:

- Sería útil destacar más claramente la contribución específica del trabajo respecto a estudios previos.
- Se menciona que el modelo tiene limitaciones, pero se podría profundizar más en por qué GraphRNN no logra replicar las propiedades estructurales de los grafos.
- ¿Por qué no se probaron otros modelos más recientes? Se mencionan alternativas, pero no se comparan experimentalmente.

---

### Decision · Program_Chairs · 2025-03-19

Accept